# Sensor-Based Evaluation of Purslane-Enriched Biscuits Using Multivariate Feature Selection and Spectral Analysis

**DOI:** 10.3390/s25247548

**Published:** 2025-12-12

**Authors:** Stanka Baycheva, Zlatin Zlatev, Neli Grozeva, Toncho Kolev, Milena Tzanova, Zornitsa Zherkova

**Affiliations:** 1Department of Food Technologies, Faculty of Technic and Technologies, Trakia University, 38 Graf Ignatiev Str., 8600 Yambol, Bulgaria; stanka.baycheva@trakia-uni.bg (S.B.); toncho.kolev@trakia-uni.bg (T.K.); 2Department of Biological Sciences, Faculty of Agriculture, Trakia University, Students’ Campus, 6000 Stara Zagora, Bulgaria; n.grozeva@trakia-uni.bg (N.G.); milena.tsanova@trakia-uni.bg (M.T.); zornitsa.zherkova.16@trakia-uni.bg (Z.Z.)

**Keywords:** food sensors, signal processing, reflectance spectroscopy, non-destructive analysis, smart food monitoring, chemometric modeling

## Abstract

This study presents a sensor-integrated framework for evaluating purslane (*Portulaca oleracea* L.) stalk flour as a functional ingredient in butter biscuits. A Design of Experiments (DoEs) approach was applied using multisensor probes (electrical conductivity, pH, TDS, ORP) and digital imaging sensors (visible reflectance spectra) for real-time, non-destructive quality assessment. Multivariate analysis with Repeated Relief Feature Selection (RReliefF) and Principal Component Analysis (PCA) reduced 54 initial measurements to 19 informative features, with the first two principal components explaining over 96% of the variance related to flour concentration. Regression modeling combined with linear programming identified an optimal substitution level of 9.62%. Biscuits at this level showed improved texture, enhanced elemental composition (Ca, Mg, Fe, Zn), stable color, and maintained sensory acceptability. The methodology demonstrates a reliable, low-cost sensing and chemometric approach for data-driven, non-destructive quality monitoring and product optimization in food manufacturing.

## 1. Introduction

In recent decades, there has been a marked increase in interest toward functional foods enriched with plant-derived ingredients. Processing of cereals and bakery manufacturing often results in considerable losses of micronutrients, particularly in refined flour products [1], thereby creating a demand for plant-based additives that improve nutritional and physiological properties [2]. Parallel to these developments, advances in sensor technologies and chemometric modeling have facilitated precise, rapid, and non-destructive methods for food quality evaluation. Multisensor systems combining electrochemical, optical, and spectral modalities capture complex interactions among raw materials, processing conditions, and product attributes, thereby supporting a transition from empirical formulation to data-driven optimization. Despite these advances, their application to functional bakery products remains fragmented. The present study addresses this research gap by applying a multisensor framework to investigate the effects of purslane (*Portulaca oleracea* L.) stalk flour in butter biscuits, integrating electrical, optical, and spectral analyses with multivariate feature extraction and Principal Component Analysis (PCA) for modeling dough structure and final product characteristics.

Purslane (*Portulaca oleracea* L.) is a widely distributed herbaceous plant rich in omega-3 fatty acids, antioxidants, and other biologically active compounds [3,4]. It is considered a promising raw material for functional foods due to its high content of proteins, carbohydrates, cellulose, and lipids, as well as vitamins A, C, and E, and essential macro- and microelements [5,6,7,8,9]. Of particular relevance is its high concentration of α-linolenic acid and other omega-3 fatty acids, which contribute to cardiovascular health [10,11]. Phenolics, flavonoids, alkaloids, saponins, and anthocyanins account for its antioxidant and anti-inflammatory activity [12,13]. The plant further contains homoisoflavonoids and feruloyl-amides with potential pharmacological effects [14], supporting its use in both medicinal and food applications [15]. The technological properties of dried purslane stems include high water-binding capacity, swelling, emulsifying, and stabilizing behavior [16,17,18,19]. Studies indicate that purslane flour improves dough plasticity and gluten elasticity, and whole-stem flour demonstrates favorable physicochemical characteristics such as low humidity and satisfactory storage stability [5]. Incorporation levels of up to 10% do not diminish sensory quality and may improve structure and color in bakery products [20,21]. Purslane has been incorporated successfully into yogurt, biscuits, and ice cream, enhancing antioxidant activity and nutritional value [22,23]. Patents for its application in preserves, mayonnaise, and spreads confirm its stabilizing potential and ability to replace synthetic additives [17]. Due to its dietary fiber and omega-3 content [2,10,11,24], purslane flour represents a valuable ingredient for improving the nutritional profile of staple foods.

Butter biscuits are widely consumed across age groups due to their sensory appeal and convenience. Their formulation offers opportunities for incorporating plant-based functional ingredients. Purslane flour contains polyphenols, flavonoids, omega-3 fatty acids, and fiber [25], shown to enhance antioxidant potential and mineral composition [26]. Typical substitution levels range from 2 to 10%; higher additions may weaken the gluten network, reduce volume, and introduce bitterness [22]. Moderate levels (~5%) improve color, texture, and aroma, whereas excessive concentrations may diminish consumer acceptance [27]. Nevertheless, standardization of moisture content, particle size, and oxalate levels remains necessary for technological implementation [5,28].

Although numerous studies describe the properties of purslane as a raw material, fewer provide comprehensive analyses of biscuits containing purslane. Concurrently, recent developments highlight the increasing role of sensor-based quality assurance in bakery systems, including IoT-enhanced control of bread quality [20], odor-based freshness monitoring [21], computer vision and deep learning for biscuit classification [22], and multisensor platforms for food safety and quality monitoring [23]. These studies underscore the relevance of integrated sensing approaches. The objective of the present study is to develop and apply a low-cost multisensor and chemometric framework to optimize butter biscuit formulations enriched with whole purslane stalk flour and to determine incorporation levels that enhance nutritional quality while maintaining desirable technological and sensory properties. Unlike most prior work focusing on leaves or cultivated varieties, this study utilizes the entire above-ground biomass of wild *Portulaca oleracea* L. and employs a multifactorial evaluation combining physicochemical, geometric, spectral, colorimetric, and sensory analyses. The integration of multisensor analytics with feature selection enables quantitative modeling of the effects of additive concentration on dough and biscuit properties, providing a foundation for sustainable ingredient innovation and automated quality control in bakery formulations.

## 2. Materials and Methods

### 2.1. Sensor-Based Framework

A sensor-based framework was developed to enable rapid, non-destructive, and data-driven assessment of biscuit quality, ensuring consistency, safety, and enhanced nutritional value through advanced sensor technologies and chemometric analysis.

Table 1 presents a sensor-based and chemometric framework designed to assess the quality of purslane-enriched butter biscuits. The process begins with sensor data acquisition, where spectral sensors (such as VIS spectrophotometers and video sensors) capture reflectance data for rapid, non-destructive analysis, while basic chemical sensors measure pH and electrical conductivity to evaluate acidity and ionic content. In the data processing and modeling stage, software tools like Matlab and Excel are used to preprocess the data by reducing noise and correcting artifacts, followed by multivariate analysis using PCA to select relevant features and build predictive models. The final stage involves evaluation and interpretation, where statistical visualizations (e.g., score and loading plots) illustrate sample groupings and model accuracy, and scientific reasoning is applied to interpret performance metrics such as R^2^, standard error, and *p*-values, validating the framework’s effectiveness and potential for broader application.

### 2.2. Collection and Analysis of Data for Optimizing the Technical Procedure

The analysis was conducted in two stages. First, sensor-based measurements were obtained from real biscuit samples enriched with purslane stalk flour. Second, these measurements were compared with equivalent features reported in the literature. This sequence ensured that our experimental data were grounded in established ranges while allowing direct comparison on overlapping features. Differences in additional features (e.g., spectral indices) reflect the novelty of our sensor-based framework and the use of whole stalk flour, which extends beyond prior studies focused mainly on leaves or cultivated varieties.

The results from both stages were subsequently compared with findings from other literature sources. The experimental design followed a structured DoE approach, where purslane flour concentration was treated as the main factor (0–15%), and sensor-derived variables (pH, EC, TDS, ORP, reflectance spectra) were systematically evaluated. This ensured balanced sampling, minimized bias, and allowed robust statistical interpretation of multivariate data.

### 2.3. Purslane Supplements

The purslane supplement is produced from finely ground whole stalks, preserving the plant’s essential nutrients and bioactive compounds. Stalks were harvested during the 2024 growing season from natural populations in the Thracian Plain. Immediately after collection, the stalks were carefully rinsed under running water to remove impurities and debris. They were then rapidly dried at 40 °C and ground into a fine powder. This process ensures minimal nutrient loss while maintaining the plant’s natural aroma and flavor.

Table A1 (Appendix A) presents the chemical composition of wild purslane from the Thracian Lowland. Data on chemical composition are adapted from Zherkova et al. [24].

### 2.4. Biscuit Receipt and Technological Parameters

The biscuits were prepared using wheat flour type 500 (Topaz Mel, Ltd., Sofia, Bulgaria), iodized table salt, butter “Sayana” (Milky Group Ltd., Haskovo, Bulgaria), and chicken egg yolks (in accordance with Ordinance № 1 of 9 January 2008 on the requirements for marketing table eggs).

The chemical composition of wheat flour type 500 (Topaz Mel, Ltd., Sofia, Bulgaria) was determined according to standard AOAC methods. The flour contained approximately 11.2% protein, 1.3% fat, 0.5% crude fiber, 0.55% ash, 75.0% carbohydrates, and 11.5% moisture.

Table 2 shows the formulation and composition of the biscuit samples with the additive. The amounts of wheat flour and the additive were adjusted, while the quantities of all other ingredients remained constant.

Table A2 (Appendix A) outlines the procedure for preparing butter biscuits with the additive, which consists of 11 steps. Some operations are performed at room temperature, while those requiring the butter to remain firm are carried out under refrigerated conditions. The total preparation time using this method is approximately 2.5–3 h.

### 2.5. Analytical Methods

Although multiple sensor parameters were recorded, the optimization process was restricted to purslane flour concentration. Other variables were used only to support feature selection and model validation, thereby ensuring practical relevance.

#### 2.5.1. Determination of Physicochemical Characteristics

Samples for the determination of active acidity (pH), electrical conductivity (EC), total dissolved solids (TDSs), and oxidation-reduction potential (ORP) were prepared following AACC Method 02-52.01 [29], “Hydrogen-Ion Activity (pH)—Electrometric Method.” [30] According to the procedure, distilled water was heated to 70 °C, and the sample was dissolved in a 1:10 ratio (5 g of sample in 50 mL of distilled water). The mixture was stirred until a homogeneous solution was obtained. After cooling to room temperature, three consecutive measurements were taken for each parameter, and the mean values and standard deviations were calculated.

The quantities of ingredients were measured using a Pocket Scale MH-200 (ZheZhong Weighing Apparatus Factory, Yongkang, China) with a maximum capacity of 200 g and a resolution of 0.02 g.

Active acidity (pH), electrical conductivity (EC, µS/cm), and total dissolved solids (TDS, ppm) were determined using a combined PH-TDS-EC-TEMP meter (Nanjing Tsung Water Technology Company Ltd., Nanjing, China). pH calibration was performed using a three-point calibration (pH 4.00, 7.00, and 10.00) with NIST-traceable buffer solutions (Hanna Instruments, Woonsocket, RI, USA) at 25 ± 1 °C. Electrical conductivity (EC) and TDS sensors were calibrated using a standard 1413 µS/cm potassium chloride (KCl) solution, verified with a certified reference conductivity meter (±1.5% deviation). EC measurements are sensitive to sample preparation and matrix heterogeneity. Purslane flour contains variable elemental fractions, and its distribution in dough can influence ionic release, contributing to higher uncertainty compared to pH or ORP.

The oxidation–reduction potential (ORP) of the aqueous extract was measured potentiometrically. Sample preparation followed the AACC Method 02-52.01 [29]. A platinum electrode with an Ag/AgCl reference electrode (Model ORP-2069, Shanghai Longway, Shanghai, China) was used for the ORP measurement in mV. The oxidation–reduction potential (ORP) probe was standardized against a 470 mV Zobell solution (Reagecon Diagnostics, Shannon, Ireland) to ensure accurate redox potential readings.

Temperature (T, °C) was recorded using a digital thermometer V&A VA6502 (Shanghai Vihua V&A Instrument Co., Ltd., Shanghai, China).

The parameters pH, EC, ORP, and TDS are estimated in the purslane raw material, biscuit dough, and baked biscuits. All sensor calibrations were performed immediately before measurements and repeated every 20 samples or every 2 h of operation, whichever occurred first. Sensor drift was monitored, and recalibration was performed when the deviation exceeded 2%.

#### 2.5.2. Determination of Thermal Losses

Baking losses were determined by cooling the baked biscuit samples to room temperature and measuring their mass on a technical scale with 0.1 g accuracy. A technical scale, Boeco BBL-64 (Boeckel GmbH, Hamburg, Germany), with a maximum capacity of 300 g and an accuracy of 0.1 g, was used. Thermal losses were calculated using the following formula:(1)$$TL=\frac{a-b}{a}\times 100\%$$
where *a* represents the mass of the sample before baking (g) and *b* the mass of the sample after baking (g).

#### 2.5.3. Determination of Elemental Composition

The elemental composition of the biscuits enriched with purslane flour was determined at the licensed laboratory “Scientific Research Laboratory Complex” (Trakia University, Stara Zagora, Bulgaria), with test report 32/28 April 2025. Measurements were performed using an Atomic Absorption Spectrophotometer, model AAnalyst 800 (PerkinElmer, U.S. LLC, Shelton, CT, USA), following laboratory methods VVLM 2/2024 and VVLM 6/2024.

Chemical analysis of the biscuit samples was conducted in the same laboratory (report 22/8 April 2025), according to laboratory procedure VVLM 3/2024 and the following standards: BDS EN ISO 5983-1:2006 [31]—Animal feeding stuffs—Determination of nitrogen content and calculation of crude protein—Part 1: Kjeldahl method; BDS ISO 6492:2007 [32]—Animal feeding stuffs—Determination of fat content; BDS ISO 6496:2000 [33]—Animal feeding stuffs—Determination of moisture and other volatile matter; BDS ISO 5984:2022 [34]—Animal feeding stuffs—Determination of crude ash content.

#### 2.5.4. Determination of Spread Factor

The spread factor (*SF*) was determined by measuring the diameter (*D*, mm) and height (*h*, mm) of the biscuits after baking. The diameter was measured at three points, and the average value was calculated, while the height was measured at three points and averaged in the same manner. Measurements were taken using a digital caliper SEB-DC-023 (Shanghai Shangerbo Import & Export Co., Ltd., Shanghai, China) with an accuracy of 0.05 mm and a maximum measuring length of 150 mm. The spread factor was then calculated using the following formula:(2)$$SF=\frac{D}{h}$$
here *D* is the diameter of the biscuit after baking (mm) and *h* is the height of the biscuit after baking (mm). The resulting value is a dimensionless quantity.

#### 2.5.5. Sensory Evaluation of Biscuits

Sensory analysis of the biscuits was conducted by a trained nine-person panel (five academic staff, four graduate students) from the Food Technologies Department, following the methodology of Bulgarian State Standard (BDS) EN ISO 13299:2016 [35] (Sensorial analysis).

Tasting took place in a controlled environment using individual booths, standardized D65 lighting to prevent visual bias, and a randomized, three-digit coding system to ensure impartial evaluation. Panelists cleansed their palates with water between samples.

The butter biscuits evaluated were compliant with the Bulgarian national standard BNS 441:1987 [36] (Biscuits. General requirements), and the pigweed ingredient was food-grade as per Bulgarian Regulation No. 5/2018 on organic production. All applicable ethical and regulatory requirements were met, as sensory analysis of food products in Bulgaria (Regulations No. 2/2024 and No. 12/2021) does not require ethical approval when human or psychological factors are not involved.

The sensory evaluation parameters are General appearance, Consistency, Aroma, Taste, Smell, Chewiness, and Overall evaluation.

Biscuits were rated using a 5-point Likert scale (1 = completely does not correspond to the indicator; 5 = completely corresponds to the indicator), and an overall average organoleptic score was calculated.

#### 2.5.6. Obtaining Color Digital Images

Color digital images were captured using the video sensor of an LG L70 mobile phone (LG Electronics, Inc., Seoul, Republic of Korea). The sensor, model VB6955CM (STMicroelectronics International N.V., Geneva, Switzerland), provides a resolution of 2600 × 1952 pixels with a pixel size of 1.4 × 1.4 μm.

Spectral and color data obtained via the integrated video sensor were calibrated using a white reference tile (L = 94.82, a = 1.05, b = −0.35) and a black standard (L = 0) prior to each image acquisition sequence. The calibration ensured consistent illumination and color accuracy within ±1 ΔE across the imaging set.

Images were initially obtained in the RGB color model and then converted to the Lab color model according to CIE Lab (1976), using standard conversion functions for a 2° observer and D65 illumination.

#### 2.5.7. Calculation of Color Difference

The color difference (ΔE) was calculated, with values ranging from 0 to 100. Values near 0 indicate that the color of the biscuits with the additive is very similar to that of the control sample, while values closer to 100 reflect bigger differences. A ΔE value below 20 is generally difficult to perceive with the naked eye.(3)$$∆E=\sqrt{{\left(L_{c}-L_{a}\right)}^{2}+{\left(a_{c}-a_{a}\right)}^{2}+{\left(b_{c}-b_{a}\right)}^{2}}$$
where *L_c_*, *a_c_*, and *b_c_* are the color components of the control sample, and *L_a_*, *a_a_*, and *b_a_* are the color components of the sample with the additive.

#### 2.5.8. Calculation of Color Indices

The following color indices were calculated: C1 represents how light and brown the color is. Brown shades typically have moderate to low L, slightly positive a, and moderately high b. C2 indicates how dark and brown the color is, with greater emphasis on lower lightness. C3 reflects how yellow-brown the color is, emphasizing the yellow component while retaining the brown tone. C4 measures the richness of the brown color, highlighting very dark brown shades.

The color indices were calculated using the following formulas:(4)$$C_{1}=0.5\times L+\left(50-a\right)+\left(b-5\right)$$(5)$$C_{2}=\left(100-L\right)+2\times b-0.5\times a$$(6)$$C_{3}=b+\left(100-L\right)-a$$(7)$$C_{4}=2\times \left(100-L\right)+b-a$$

#### 2.5.9. Obtaining Spectral Characteristics and Calculation of Spectral Indices

Spectral characteristics were obtained by converting values from the XYZ and LMS color models into reflectance spectra in the visible (VIS) range (390–730 nm), using the mathematical formulas described by Vilaseca et al. [37]. The calculations were based on a 2° standard observer and D65 illumination. Reflectance spectra were recorded in the 390–730 nm range with a 1 nm step size. While ISO standards typically specify 400–700 nm with 10 nm increments, our extended range and finer resolution allow more detailed spectral characterization. For comparison, data can be aggregated to 10 nm intervals to align with the ISO methodology.

The following spectral indices were calculated: S1 indicates the balance between red and green, useful for distinguishing reddish-brown from greenish-yellow colors; S2 measures the slope of the spectral curve between two wavelengths, reflecting the dominance of specific color transitions and aiding in the analysis of color changes across the spectrum; S3 assesses the depth of a reflectance feature, which helps identify colors with strong absorption in particular spectral regions.

The spectral indices were determined at two wavelengths, 530 nm and 630 nm, using the following formulas:(8)$$S_{1}=\frac{R_{630}}{R_{530}}$$(9)$$S_{2}=\frac{R_{530}-R_{630}}{20}$$(10)$$S_{3}=\frac{R_{530}-R_{630}}{R_{530}+R_{630}}$$

#### 2.5.10. Feature Vectors and Selection of Informative Features

Feature vectors were constructed to describe the flour blends, dough, and biscuits. The features and their significance are presented in Table 3. Features F1 to F4 correspond to flour, F5 to F16 to dough, and F17 to F54 to biscuits.

Informative features were selected using the RReliefF method. ReliefF captures relationships between features and is robust to noise. Features with weight coefficients above 0.6 were considered informative [38], and a vector of these selected features was defined.

#### 2.5.11. Data Reduction Method

The feature vector data were then reduced using PCA [39]. PCA is a statistical method for dimensionality reduction that efficiently handles large datasets by transforming them into a smaller set of variables called principal components. Each principal component is a linear combination of the original variables, ordered by the amount of variance it explains.

The analysis combines the percentage of additives with the selected informative features. This was performed for all additive levels used (0–15%), considering only the informative features.

Although Principal Component Regression (PCR) is a widely used dimensionality reduction technique, it was not applied in this study because PCR constructs components solely based on predictor variance, without considering their relevance to the response variable. This can reduce interpretability in food formulation studies where the target attributes are specific product quality indicators. Instead, we employed the Relief Feature Selection for Regression (RReliefF) algorithm to identify variables directly associated with biscuit quality, followed by PCA for dimensionality reduction. This two-step approach ensures that only informative features are retained, improving robustness in noisy, multivariate datasets and enhancing predictive accuracy [40].

#### 2.5.12. Regression Model and a Linear Programming Algorithm

A regression model, frequently used in the analysis of food products [41], was applied. In Equation (11), the variable ‘z’ denotes the percentage of purslane flour incorporated into the biscuit formulation. The equation describes the relationship between the independent and dependent variables and is expressed as follows:(11)$$z=b_{0}+b_{1}x+b_{2}y+b_{3}x^{2}+b_{4}xy+b_{5}y^{2}$$
where *z* is the dependent variable, *x* and *y* are the independent variables, and the model coefficients are denoted as *b*.

The model was evaluated based on the coefficient of determination (R^2^), the model coefficients, their standard errors (SE), *p*-values, and the Fisher criterion (F). A residual analysis was also performed.

To determine the optimal amount of additive, a linear programming algorithm was used, implemented via the “linprog” built-in function in Matlab 2017b program environment. In linear programming, the task is to find a vector x that maximizes or minimizes a linear function f^T^x, subject to linear constraints:(12)$$\underset{x}{\min}\;f^{T}x$$
so that one of the conditions is satisfied:(13)$$Ax\leq b\;\;\;\;\;\;\;A_{eq}x=b_{eq}\;\;\;\;\;\;\;\;l\leq x\leq u$$
where *f*, *x*, *b*, *beq*, *l*, and *u* represent vectors, and *A* and *A_eq_* represent matrices.

An “Interior-point-legacy” algorithm was used (built-in function in Matlab 2017b program environment). This algorithm finds an optimal solution by traversing the interior of the feasible region.

### 2.6. Statistical Analysis

The data were statistically processed with Matlab 2017b (The Mathworks Inc., Natick, MA, USA). Preparation and visualization of data were performed with MS Office 2016 (Microsoft Corp., Washington, DC, USA).

All measurements were made in three independent repeats, and the means are used in the tables and graphs. The statistical functions standard deviation and maximum allowable error were used. All data were processed at an accepted significance level of α < 0.05.

One-way ANOVA was used to analyze the data. A post hoc LSD test was also performed to determine the degree of significance of differences between average values after statistically significant differences (*p* < 0.05) were found. The non-parametric Kruskal–Wallis test was used when there was no normal distribution.

All statistical analyses were performed at a significance level α = 0.05. Calculated *p*-values are reported, and differences were considered statistically significant when *p* < α (0.05).

### 2.7. Literature Data

Table 4 presents the characteristics summarized from the literature (FLx) for purslane biscuits and their significance. The first group, FL1 to FL7, includes organoleptic properties. The second group covers antioxidant activity (FL8) and the chemical properties of the product. In addition, the hardness of the final product (FL13) was also considered.

Table A4 (Appendix B) summarizes the values of the analyzed characteristics of purslane biscuits reported in the literature [5,42,43,44]. The data cover additive levels ranging from 0% to 30%.

## 3. Results

### 3.1. Preparation and Analysis of Real Biscuits

#### 3.1.1. Analysis of the Raw Material

Table 5 shows that purslane exhibits an acidic pH of approximately 5.56. TDS is around 2652 ppm, and EC is approximately 5355 µS/cm, indicating a high concentration of dissolved substances. The oxidation-reduction potential (ORP) is about 147 mV, reflecting the characteristic oxidative properties of the raw material.

#### 3.1.2. Flour Analysis

Table 6 summarizes the physicochemical properties of flour with purslane added at levels ranging from 0% to 15%. The addition of purslane significantly alters the flour’s physicochemical characteristics. As the concentration of the additive increases, pH decreases, reaching 7.35 at 15% addition. Higher purslane levels also lead to increased TDS and EC, with the highest values observed at 15%. The ORP remains relatively stable, ranging between 40 and 47 mV.

#### 3.1.3. Dough Analysis

Table 7 shows the physicochemical properties of biscuit dough with purslane added at levels of 0%, 5%, 10%, and 15%. Adding purslane results in significant changes in these properties. The pH rises from 6.74 in the control sample (0% addition) to 7.43 at 15% addition. TDS and EC increase from 1877 ppm and 3762.5 µS/cm to 2752.5 ppm and 5519 µS/cm, respectively, indicating a higher content of dissolved substances and ions. Meanwhile, the ORP decreases from 31 mV to 21.5 mV, reflecting increased reductive activity.

Based on Figure A1 (Appendix A), the color and surface structure of the biscuits dough are affected by the addition of purslane flour. The control sample (0%) exhibits a smooth, distinctly yellowish color. As the percentage of purslane flour increases (to 5%, 10%, and 15%), the dough’s appearance undergoes a progressive change. Its color shifts from yellow to a more muted greenish-gray/brown, and the surface texture becomes noticeably more granular and rougher due to the presence of the added plant-based flour.

Figure A2 (Appendix A) visualizes the Lab and Spectral characteristics of biscuit dough with purslane. In the Lab color space, increasing the amount of purslane leads to a decrease in lightness (L), making the dough darker. The “a” and “b” axes, representing the red-green and yellow-blue components, show different shades: at 0% addition, the color is more neutral, while at 15% addition, it takes on greener and more yellowish tones. Spectral analysis of diffuse reflectance indicates that higher levels of addition result in changes in the spectral peaks, lower reflectance in the blue region, and increased absorption in the red region of the light spectrum. This suggests that purslane affects the light absorption of the dough, which may be related to changes in composition and water content. At 5% substitution, a larger scatter was observed. This reflects the transitional state of the dough matrix, where gluten structure and purslane-derived fiber/elemental interact, making sensor readings more sensitive to small compositional differences.

Figure 1 shows the color difference ΔE relative to the control biscuit dough sample, depending on the amount of additive purslane. As the additive level increases, the color difference also increases, reaching values that make the color visibly distinguishable to the naked eye.

Table 8 presents the values of the color indices of biscuit dough with varying percentages of the additive. As the concentration increases from 5% to 15%, the values of C1, C2, and C3 decrease, indicating a deepening of color and reduced lightness of the dough. The lower C2 value at 15% addition (57.7) compared to the control (144.07) suggests enhanced yellowish and greenish tones, characteristic of purslane. In contrast, C4 shows smaller variations, with the highest values observed at 5% addition, which may be associated with a more balanced spectral profile of the dough.

Table 9 presents the values of the spectral indices of biscuit dough with varying percentages of additive. The indices of dough with purslane show significant changes as the percentage of the additive increases. S1 decreases from 0.66 in the control sample to 0.57 at 10% addition, suggesting reduced light reflectance and potential changes in dough structure. S2 shows a marked increase from 0.51 at 0% addition to 1.16 at 15% addition, indicating higher pigment content and enhanced absorption properties of the dough. S3 increases slightly with the addition of purslane, stabilizing at 10% and 15%, which suggests a consistent spectral characteristic of the dough.

#### 3.1.4. Biscuit Analysis

Table 10 presents the elemental composition of biscuits with purslane. Enrichment with purslane flour leads to significant improvements compared to the control sample (0% addition). As the purslane content increases (5%, 10%, 15%), Ca rises from 950.53 mg/kg to 1523.51 mg/kg, enhancing the elemental profile of the product. The content of K rises significantly, reaching 5187.78 mg/kg at 15% addition, potentially enhancing nutritional value. Levels of Fe, Mg, and Zn are similarly enhanced, supporting the functional and health-promoting properties of the biscuits. For instance, Mg nearly doubles at 15% addition (764.46 mg/kg compared to 287.97 mg/kg in the control), potentially benefiting metabolic processes. A steady increase is observed in Cu and Mn, reaching 5.91 mg/kg and 8.73 mg/kg, respectively, which may contribute to stronger antioxidant activity. The levels of P remain relatively stable at 0.15–0.17%, indicating a moderate effect of the additive on this element.

Table 11 presents the results of the chemical analysis of biscuits made with purslane flour. Moisture decreases with increasing purslane content, reaching 3.29% at 15% addition, indicating a drier and more stable product. Dry matter increases from 96.09% to 96.71%, also reflecting reduced water content and improved stability. Crude protein content rises slightly at 10% addition (10.72%) but decreases again at 15% (10.51%), suggesting a limit to the beneficial interaction with purslane. Crude fat ranges from 30.93% to 32.44%, with the highest value observed at 10% addition.

A notable effect is seen in crude fiber, which increases from 19.79% in the control sample to 23.77% at 15% addition, enhancing the functional properties of the biscuits and supporting better digestion. Crude ash also rises from 1.33% to 3.85%, indicating increased elemental content. Extractive nitrogen substances (ENSs) decrease with higher additive levels (from 33.81% to 27.11%), reflecting a lower content of easily degradable carbohydrates, which may contribute to a more balanced glycemic index of the product.

Although purslane flour is characterized by high water absorption capacity, the biscuits prepared with purslane showed lower final moisture content. This apparent contradiction is explained by the difference between flour hydration in suspension and moisture retention in baked products. The higher fiber and mineral content of purslane increases water binding initially, but during baking, it promotes faster evaporation and weakens the gluten network’s ability to retain water. Consequently, biscuits with purslane flour exhibit reduced moisture compared to the control.

Purslane flour contains more fat than wheat flour (4.25% vs. 1–1.5%), and biscuits with 15% purslane showed lower fat content. This apparent paradox is explained by dilution effects from increased fiber and mineral fractions, oxidation of unsaturated lipids during baking, and matrix heterogeneity at higher substitution levels. Therefore, the measured reduction reflects processing and compositional interactions rather than inconsistency in raw material composition.

The main physicochemical characteristics of the biscuits are presented in Table 12. The addition of purslane leads to significant changes in the biscuits’ physicochemical properties. As the concentration of the additive increases, the pH rises, reaching 7.37 at 15%, indicating a more acidic environment. The treatment also increases TDS, EC, and ORP, with the highest values observed at 15% addition. This indicates that increasing the amount of purslane affects the chemical properties of the biscuits, altering their acidity, conductivity, and oxidative status, with all these differences being statistically significant.

Table 13 presents the changes in the geometric properties and heat losses of biscuits during baking. The addition of purslane flour significantly affects both the physical and thermal characteristics of the biscuits. With increasing additive levels, the diameter and spread factor increase, while the height decreases, indicating that the biscuits become flatter and more expanded. Heat losses also rise slightly at higher addition levels, suggesting a greater tendency for thermal loss during baking. All these changes are statistically significant, highlighting the influence of purslane on the biscuits’ physical and thermal properties.

Table 14 presents the results of the sensory evaluation of biscuits with varying percentages of additives. Adding purslane flour at levels above 10% negatively affects the organoleptic properties of the biscuits, reducing scores for appearance, texture, aroma, taste, smell, and chewiness. All these differences are statistically significant, confirming that increasing the additive concentration substantially alters the sensory characteristics of the product.

Color digital images of the resulting biscuits are shown in Figure A3 (Appendix A). As the amount of additive increases, both the color characteristics and the surface structure of the product are noticeably altered.

Figure A4 (Appendix A) visualizes the Lab and spectral characteristics of biscuits with purslane. It is evident that samples with higher concentrations (e.g., 15%) are grouped separately, with the overall trend showing an increase in color component values as the additive level rises. As the amount of additive increases, the spectral curve shifts and changes, with observed decreases or increases in reflectance values within specific ranges. The addition of purslane has a significant effect on both the color and spectral characteristics of the biscuits, with these changes being clearly visible and amenable to quantitative analysis.

The observed color changes in purslane-enriched biscuits are partly due to pigments (chlorophyll, carotenoids, and flavonoids) introduced with the flour. However, color is also influenced by Maillard browning reactions during baking, which intensify with higher protein and sugar content, and by matrix effects such as fiber and mineral content that alter light scattering. Thus, biscuit color serves as a composite indicator of both pigment incorporation and physicochemical transformations.

Figure 2 shows the color difference ΔE relative to the control biscuit sample, depending on the amount of additive. The data indicate that even low levels of additive (5%) result in a significant color difference, noticeable to the naked eye.

Table 15 presents the color index values of biscuits with varying percentages of additives. The C1 values decrease with increasing additive from 104.97 at 0% to 83.72 at 15%, indicating darkening or a color change. C2 also decreases, from 124.53 at 0% to 98.58 at 15%, with values of 118.89 and 111.81 at 5% and 10% addition, respectively. C3 increases from 75.04 at 0% to 81.45 at 10%, then decreases to 76.9 at 15%, which may reflect a change in hue or color saturation. C4 rises significantly from 118.43 at 0% to 139.79 at 10%, then slightly decreases to 135.69 at 15%.

Table 16 presents the spectral index values of biscuits with varying percentages of additives. The spectral index S1 remains almost constant across all additive levels, with a slight variation around 0.62, decreasing to 0.61 at 15% addition. Index S2 increases from 0.8 at 0% addition to 0.75 at 15%, with values of 0.64 and 0.66 at 5% and 10% addition, respectively. S3 values remain approximately constant at around 0.24 for all additive levels, showing no significant fluctuations.

### 3.2. Statistical Analysis and Determination of the Optimal Additive Amount

Figure A5 (Appendix A) presents the results of informative feature selection using the RReliefF method. Using a threshold of 0.6 for the weight coefficients, the selected features include physicochemical, geometric, color, and spectral indicators. The main physicochemical features above the threshold include moisture content, crude fiber, crude ash, and ENS, which play a role in determining the texture and stability of the biscuits. Among the geometric characteristics, the most informative are diameter, height, and spread factor, indicating a direct relationship between the purslane additive and the physical dimensions of the biscuits. Color and spectral indices are also highly expressed (>0.6), demonstrating that the additive has a significant effect on the product’s appearance.

The following vector of 40 informative features has been formed:FV = [F2 F3 F6 F7 F8 F9 F10 F11 F12 F13 F14 F15 F16 F18 F19 F20 F21 F22 F25 F28 F30 F31 F33 F34 F35 F36 F37 F38 F39 F40 F41 F42 F43 F44 F45 F46 F47 F52 F53 F54](14)

A PCA was conducted to investigate the relationship between the additive percentage and the measured features. Prior to the analysis, all data were normalized to the [0, 1] range. The PCA results are shown in Figure 3 and indicate that increasing the additive percentage significantly affects certain specific characteristics. Features F2, F3, F16, F28, and F43 show a strong association with PC1, reflecting changes corresponding to increases or decreases in additive levels. For PC1, low additive levels (0–5%) correspond to predominantly positive feature orientations, while higher levels (10–15%) exhibit greater variation, with some features shifting into negative values. PC2 is primarily associated with other features, such as F10 and F20, and shows a more complex relationship with additive content. Higher additive percentages are linked to increased variation along the principal components. Features such as F2, F3, F16, and F28 are particularly sensitive to additive changes, making them useful for modeling or optimization. These features are related to the biscuits’ TDS, EC, and the color differences between additive-containing samples and the control.

Table A3 (Appendix A) represents the loading matrix, which confirms that the first two principal components captured the dominant structure of the dataset. PC1 exhibited consistently high positive coefficients across several variables (0.32–0.43), indicating strong contributions from mineral composition and spectral indices. In contrast, PC2 showed moderate positive loadings (0.38–0.40) for color-related variables, highlighting the influence of chromatic attributes on sample differentiation. Negative loadings on PC1 were observed for redox and sensory parameters, suggesting that these features contributed less to the nutritional–spectral dimension. PC2 also displayed mixed signs, with positive contributions from color indices and negative contributions from technological attributes, reflecting the balance between visual quality and processing effects. The separation of loadings shows that PC1 primarily represents nutritional and spectral enrichment, while PC2 emphasizes colorimetric and sensory variation. This dual structure supports the interpretation that purslane incorporation simultaneously enhances mineral and spectral properties while modulating color and sensory attributes, with optimal substitution levels aligning along regions of balanced PC1 and PC2 scores.

A regression model of the form %A = f(PC1, PC2) was obtained. Non-informative coefficients with *p* > α were removed. The model equation is as follows:(15)$$\%A=9.99+0.24{{PC}_{1}}^{2}-35.02{{PC}_{2}}^{2}-23.94{PC}_{1}{PC}_{2}$$

The coefficient of determination is R^2^ = 0.99. According to the Fisher criterion, F(3, 8) = 323.72, ≫ Fcr = 4.07. The significance level is *p* ≪ 0.05, and the standard error is SE = 0.62.

The regression model predicting %A from the two principal components, PC1 and PC2, including their interactions, shows excellent performance with an R^2^ of 0.99, indicating that it explains 99% of the variation in %A. All predictors and their interactions are significant at *p* < 0.00. The terms PC1^2^ and PC2^2^ indicate that the effects of PC1 and PC2 increase or decrease with their respective values.

Residual analysis shows that the residuals lie close to a straight line on the normal probability plot and are distributed nearly normally. These results support the conclusion that the regression model accurately describes the experimental data.

The optimal amount of additive in the biscuits has been determined, as shown in Figure 4. The positive PC1 value indicates that an addition of 9.62% (noted as green asterisk on the figure) preserves certain properties closer to the lower additive levels (0%, 5%, 10%), making the biscuits more similar to the control sample. In particular, these characteristics are closely associated with heat loss, diameter, spread factor, and some sensory attributes, such as taste and aroma, of the control.

The second principal component, PC2, is near zero, indicating minimal deviation at 9.62% additive, which helps maintain critical biscuit characteristics without causing extreme changes.

### 3.3. Research and Selection of the Optimal Purslane Supplement Amount by Literature Data

Figure A6 (Appendix B) shows the results of selecting informative features using the RReliefF method. With a weight coefficient threshold of 0.6, the selected features are primarily organoleptic characteristics, along with fat and protein content.

The selected vector of seven informative features, obtained from literature data on purslane-enriched biscuits, is as follows (Equation (16)):FVL = [FL1 FL2 FL3 FL4 FL5 FL12 FL15](16)

Although the dataset contained seven features, PCA was applied to visualize correlations and variance structure among them. RReliefF ranks features by relevance to the response variable, while PCA complements this by identifying clusters and redundancies. The combined approach improves interpretability and robustness.

The data from this feature vector were normalized to the range [0, 1] and subsequently reduced to two principal components using PCA to investigate the relationship between additive levels and the selected features. The results are presented in Figure 5.

The analysis reveals a clear relationship between the additive amount and the first principal component (PC1), which is primarily influenced by feature FL15. At higher additive levels (15–30%), PC1 increases, reflecting the stronger contribution of FL15 (protein). At lower additive levels (0–10%), PC1 takes negative values, associated with higher levels of FL4 (aroma), FL12 (fat content), and FL1 (overall product appearance). The second principal component (PC2) does not show a clear dependence on additive amount and appears to reflect internal variation among FL12, FL4, and FL1.

PCA loadings of the literature dataset, presented in Table A5 (Appendix B), show three dominant groups of features. Appearance (FL1) and chewing resistance (FL2) exhibited negative loadings across PC1, corresponding to low substitution levels where sensory attributes prevailed. Fat content (FL12) showed strong positive contributions to PC2 and PC3, aligning with intermediate substitution levels (8–10%) and reflecting enhanced nutritional and colorimetric quality. Protein (FL15) was strongly associated with PC1, consistent with high substitution levels (15–30%), where mineral and nutritional enrichment dominated. The concordance between loadings and scores demonstrates that literature data support a dual optimization pathway—moderate substitution enhances sensory and color features, while higher levels affect nutritional enrichment, often at the expense of sensory acceptability.

A regression model of the form %A = f(PC1, PC2) was developed. Coefficients that were not statistically significant (*p* > α) were removed. The resulting model equation is as follows:(17)$$\%A=9.4+23.04{PC}_{1}+4.5{{PC}_{1}}^{2}-15.22{{PC}_{2}}^{2}$$

For the resulting model, the coefficient of determination is R^2^ = 0.91. According to Fisher’s criterion, F(3, 8) = 26.27 > Fcr = 4.07. The *p*-value is less than 0.001, and the standard error is SE = 3.06. Residual analysis shows that the residuals lie close to the normal probability line and are approximately normally distributed. Based on these results, it can be concluded that the regression model adequately describes the relationship observed in the experimental data.

Using the “Interior Point Legacy” method, the optimal amount of purslane additive in biscuits was determined based on literature data. The result is visualized in Figure 6. An additive level of 11.32% falls within the transition zone between low (0–10%) and high (15–30%) amounts, according to the PC1 analysis. This value lies in the positive range of PC1, indicating the beginning of an increased influence on feature FL15 (protein), typical for higher additive levels, while still exerting some effect on FL4 (aroma) and FL12 (fat content), characteristic of lower additive levels. This suggests that at 11.32% (showed as green asterisk and value), a balanced effect is achieved between the two groups of features.

### 3.4. A Comparative Analysis Between Data from Experimental (Real) Biscuits and Literature Data

The comparative PCA, presented in Table 17 of overlapping features between experimental and literature datasets, revealed consistent structural patterns. Sensory features such as appearance (F33/FL1), chewiness (F38/FL2), aroma and odor (F34/FL4), taste (F35/FL6), smell (F36/FL4), and overall acceptability (F39/FL5) exhibited negative loadings on PC1 in both datasets (ranging from –0.53 to –0.58 in the experimental data and –0.26 to –0.44 in the literature data), confirming that sensory quality decreases with increasing purslane substitution. PC2 values were mixed, with experimental data showing weak positive contributions (0.14–0.29), while literature data emphasized stronger negative values (–0.17 to –0.37), indicating a more pronounced sensory decline reported in prior studies. Nutritional parameters demonstrated divergence—protein (F46/FL15) showed negative PC1 loading in the experimental dataset (–0.31) but strong positive loading in the literature dataset (0.48), highlighting that published data point to protein enrichment at higher substitution levels. Crude fiber (F51/FL11) and moisture (F48/FL14) also aligned with nutritional enrichment, though magnitudes differed. The concordance between datasets supports a dual optimization pathway—moderate substitution (~8–10%) balances sensory and nutritional attributes, while higher levels (>15%) enhance protein and fiber content but reduce sensory acceptability.

## 4. Discussion

The results demonstrate that purslane flour significantly affects the physical, optical, and sensory characteristics of biscuits. Increasing the additive level altered texture and color indices, which is consistent with previous work on plant-based flours improving functional biscuit properties. PCA confirmed clear separation between samples, with PC1 reflecting major changes in TDS, EC, ORP, and color attributes. The combined use of PCA scores and RReliefF ranking provided reliable identification of influential features, supporting the suitability of multisensor analysis for evaluating plant-enriched bakery products.

The optimal additive level identified in this study was 9.62%, at which the product retained desirable geometric, structural, and sensory characteristics compared with the control. Although this level was determined for wheat-based biscuits and may not apply to gluten-free formulations or consumers with allergies, it was statistically robust across replicates and modalities. The value is slightly lower than the 11.32% reported in earlier research, which may be attributed to differences in raw material composition. The chemical profile of purslane is known to vary depending on environmental and agronomic factors, influencing water absorption, gluten interaction, and sensory properties. Thus, variability in optimal substitution is expected.

Compared with literature-based estimates derived from 16 parameters, the present analysis incorporated 54 measurable features (40 informative), allowing a more accurate determination of the optimal level. Whole-stalk purslane flour showed higher protein, fiber, and mineral content than wheat flour, leading to improved nutritional composition at 9.62% substitution, including higher dietary fiber (5.27%) and Ca, Mg, and Fe content, exceeding the values reported by Mastud et al. [43]. Antioxidant activity reached 83.41% RSA, corresponding to Kurbanov et al. [18], but at a lower addition level, indicating better efficiency of the raw material. The increase in hardness is attributed to the higher fiber content, which limits gluten development and increases water binding.

Organoleptic evaluation showed high acceptability at 9.62%, exceeding the results of Waleed et al. [28], who observed reduced textural scores at 10%. EC increased to 3759 µS/cm, and pH decreased moderately, exceeding values reported by Kurbanov et al. [18], likely due to the mineral content and antioxidant components influencing redox balance. The color index (C3 ≈ 105) and ΔE < 20 agreed with the findings of Zherkova et al. [24], confirming that acceptable color can be achieved at moderate substitution levels.

These observations are consistent with previous works demonstrating the nutritional and technological benefits of purslane flour in functional foods [5,23,26,27,28]. As reported by Ahmed et al. [27], substitutions improve protein, fiber, and mineral content, while the present study additionally confirms positive effects on physicochemical and sensory properties. Similar effects of plant-based additives on structural and sensory attributes were described by Salihu et al. [45] and Goubgou et al. [46], emphasizing that such fortification strategies enhance product quality.

In contrast to earlier studies with smaller datasets or fewer variables, the multisensor framework enabled a more comprehensive assessment and precise optimization. The approach confirmed that the main technological factor affecting biscuit quality is the additive concentration, rather than the simultaneous adjustment of all parameters. Purslane flour at 9.62% enhances nutritional value and antioxidant activity without compromising sensory acceptability, supporting its use in functional product development and confirming its potential as a sustainable alternative to synthetic additives.

The integration of the RReliefF feature ranking with PCA offered an interpretable and reliable analysis workflow. Although this methodology effectively identified key variables and sample clustering, future work could incorporate Principal Component Regression, Partial Least Squares, or machine learning classifiers to further improve predictive accuracy and interpretation.

## 5. Conclusions

This study demonstrates that combined pH, EC, and spectral sensors with multivariate analysis create a reliable framework for automatic quality monitoring in bakery manufacturing. Multisensor measurements, RReliefF feature selection, and PCA models enable robust, data-driven analysis of biscuit properties. Experiment results identified an optimal purslane flour substitution level of 9.62%, which proved more precise than the 11.32% given in the literature. At this level, purslane flour improved dough consistency, enhanced texture and sensory quality, and increased nutrient content, and it maintained sensory acceptability, while still preserving biscuit diameter, color stability, and sensory acceptability. PCA confirmed that these phenomena can be explained with up to 96% accuracy, highlighting the high information content of the experimental data itself.

These findings offer a robust foundation for enhancing biscuit formulations and demonstrate the potential application of multisensor and chemometric methods in functional food innovation. The methodology may be adapted to edible sensors and for automated quality monitoring systems with IoT. It can also be applied to other foods that need constant quality monitoring. R&D investment should be made in the near future to embed multispectral sensors on the production line for real-time process control, develop gluten-free formulations, etc., as well as allergen safety tests. Future work must include detailed nutritional calculations to confirm and support specific enrichment claims, or detailed nutritional calculations are required in future work to substantiate specific enrichment claims. Comparative studies across different purslane varieties and environments in which it grows will also be effective for expanding the range of ‘purslane-enriched’ biscuits in the functional food market.

## Figures and Tables

**Figure 1 sensors-25-07548-f001:**
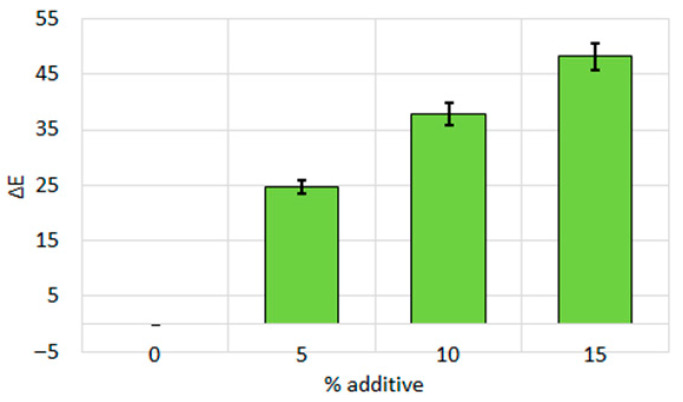
Color difference for biscuit dough between the control sample and those with different percentages of additive purslane. All data have a statistically significant difference at *p* < 0.05.

**Figure 2 sensors-25-07548-f002:**
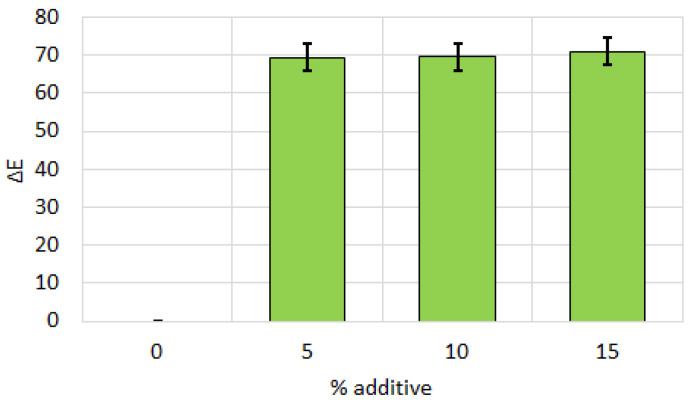
Color difference for biscuits between the control sample and those with different percentages of additive purslane flour. All data have statistically significant differences at *p* < 0.05.

**Figure 3 sensors-25-07548-f003:**
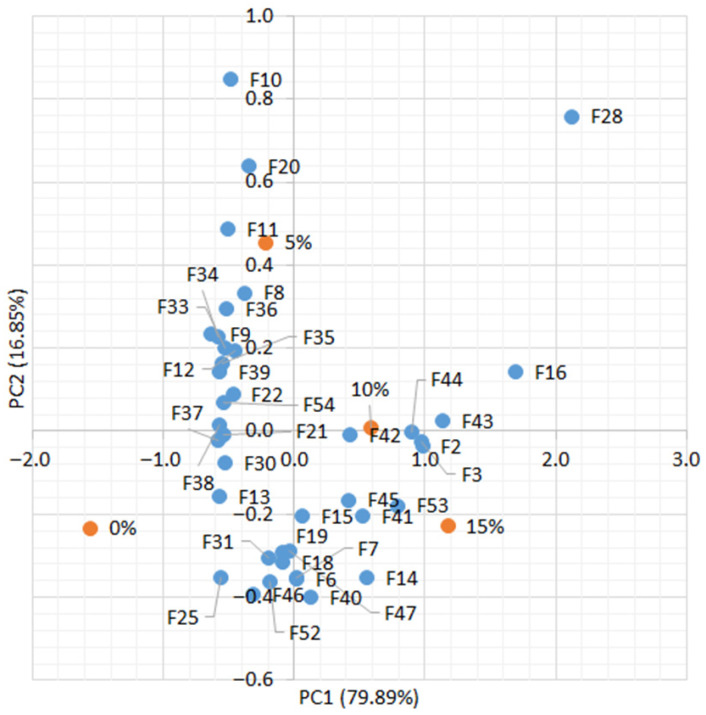
PCA of connections % additive/feature.

**Figure 4 sensors-25-07548-f004:**
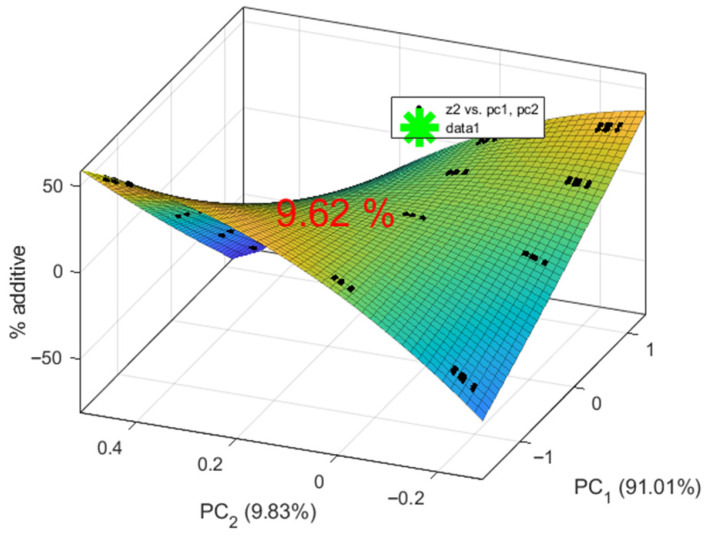
Appropriate amount of purslane additive in biscuits.

**Figure 5 sensors-25-07548-f005:**
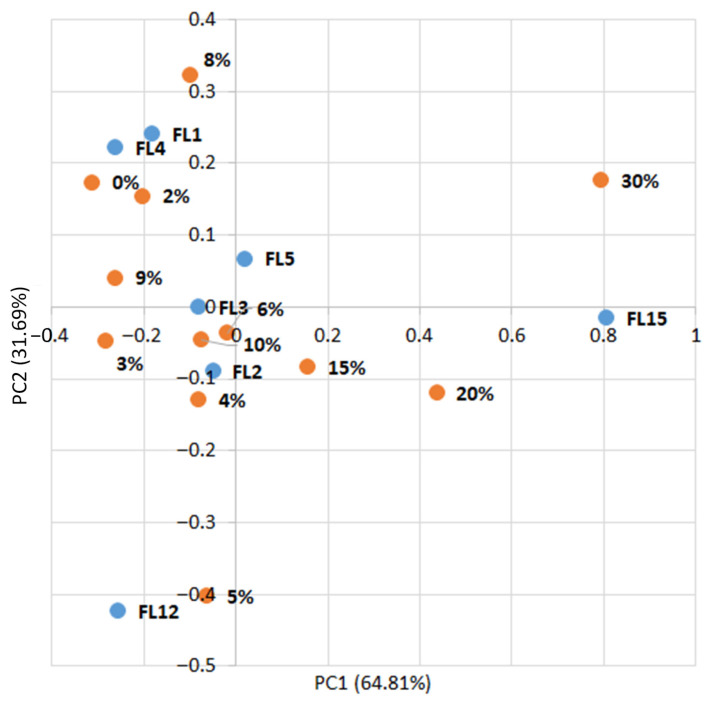
PCA of data for purslane biscuits from literature sources.

**Figure 6 sensors-25-07548-f006:**
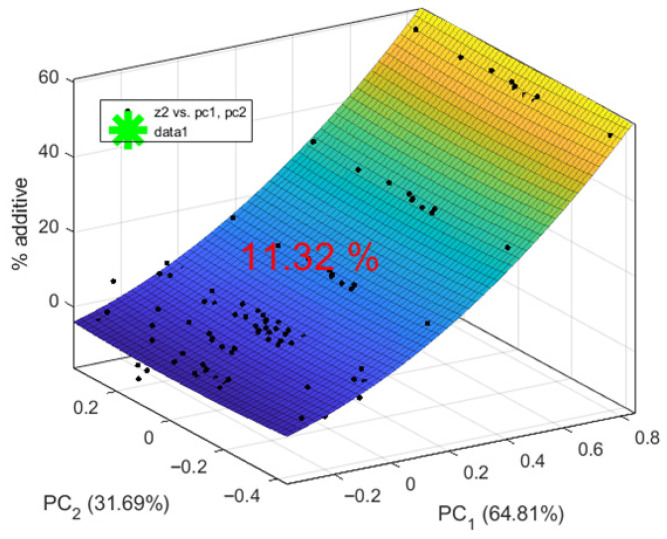
Determination of the appropriate amount of purslane in biscuits by using literature sources data.

**Table 1 sensors-25-07548-t001:** Sensor-based and chemometric framework for quality assessment of purslane-enriched biscuits.

Framework Stage	Sensor	Measurement (The Data)	Purpose in Study
1. Sensor Data Acquisition			
1.1. Spectral Sensor	VIS Spectrophotometer, Video Sensor	Spectral Signatures (Reflectance data across a specified wavelength range, e.g., 390–730 nm)	Rapid, non-destructive measurement of biscuit composition.
1.2. Basic Chemical Sensors	pH Meter with electrode probe	pH Value	Objective measurement of biscuit acidity, which affects shelf-life, texture, and flavor.
	Electrical Conductivity (EC) Meter with probe	Electrical Conductivity µS/cm	Measures ionic content; can correlate with overall structural changes.
2. Data Processing and Modeling			
2.1. Preprocessing	Data Processing Software (Matlab, MS Excel)	Noise Reduction (Smoothing, baseline correction, scatter correction)	Removes physical artifacts from raw spectral data to enhance chemical information.
2.2. Multivariate Analysis (The Framework’s Core)	Machine Learning/Chemometric Algorithms (PCA)	Feature Selection and Model Creation (PCA loadings)	Identifies the most relevant spectral and chemical features that correlate with a target property.
3. Evaluation and Interpretation			
3.1. Results Visualization	Statistical Figures	Score Plots, Loading Plots, Regression curves	Visually demonstrates sample grouping, spectral regions, and the predictive accuracy of the final model.
3.2. Final Interpretation	Scientific Reasoning	Performance Metrics: R^2^, SE, *p*-value, F-criteria	Validates the effectiveness of the sensor-based approach and discusses its potential for future application.

**Table 2 sensors-25-07548-t002:** Quantities of raw materials.

	% Additive	0	5	10	15
Raw Material					
Wheat flour, g		100	95	90	85
Purslane flour, g		0	5	10	15
Cow butter, g		50	50	50	50
Salt, g		1	1	1	1
Egg yolk, g		10	10	10	10

**Table 3 sensors-25-07548-t003:** Features and their meanings.

P	№	C	P	№	C	P	№	C	P	№	C	P	№	C	P	№	C
F	F1	pH	D	F10	C2	B	F19	EC	B	F28	dE	B	F37	S	B	F46	P
F	F2	TDS	D	F11	C3	B	F20	ORP	B	F29	D	B	F38	Ch	B	F47	Zn
F	F3	EC	D	F12	C4	B	F21	C1	B	F30	H	B	F39	OA	B	F48	M
F	F4	ORP	D	F13	S1	B	F22	C2	B	F31	SF	B	F40	Ca	B	F49	DM
D	F5	pH	D	F14	S2	B	F23	C3	B	F32	TL	B	F41	Cu	B	F50	CP
D	F6	TDS	D	F15	S3	B	F24	C4	B	F33	GA	B	F42	Fe	B	F51	CF
D	F7	EC	D	F16	dE	B	F25	S1	B	F34	C	B	F43	K	B	F52	CFB
D	F8	ORP	B	F17	pH	B	F26	S2	B	F35	A	B	F44	Mg	B	F53	CA
D	F9	C1	B	F18	TDS	B	F27	S3	B	F36	T	B	F45	Mn	B	F54	NNE

P—process; C—characteristic; F—flour; D—dough; B—biscuits; TDS—total dissolved solids; EC—electrical conductivity; ORP—oxidation—reduction potential; dE—color difference; D—diameter; SF—spread factor; TL—thermal losses; GA—general appearance; A—aroma; T—taste; S—smell; Ch—chewiness; OA—overall acceptance.

**Table 4 sensors-25-07548-t004:** Features of purslane biscuits according to literature data and their meanings.

Feature	Meaning	Feature	Meaning
FL1	Appearance	FL9	Ash, %
FL2	Chewing Resistance	FL10	Carbohydrate, %
FL3	Color	FL11	Crude Fiber, %
FL4	Odor	FL12	Fat, %
FL5	Overall Acceptability	FL13	Hardness, kg
FL6	Taste	FL14	Moisture, %
FL7	Texture	FL15	Protein, %
FL8	Antioxidant Activity, %RSA	FL16	Total Phenolic Compound, mgGAE/100 g

FL—feature obtained from literature data.

**Table 5 sensors-25-07548-t005:** Main physicochemical characteristics of purslane. All data have a statistically significant difference at *p* < 0.05.

Characteristic	Mean ± SD
Ph	5.56 ± 0.08
TDS, ppm	2651.5 ± 43.5
EC, µS/cm	5354.5 ± 93.5
ORP, Mv	146.5 ± 12.5

**Table 6 sensors-25-07548-t006:** Main physicochemical characteristics of wheat flour with purslane. All data have a statistically significant difference at *p* < 0.05.

	% Additive	0	5	10	15
Characteristic					
pH		7.45 ± 0.02	7.52 ± 0.02	7.37 ± 0.03	7.35 ± 0.04
TDS, ppm		602 ± 10	1245.5 ± 42.5	1605.5 ± 391.5	1901 ± 687
EC, µS/cm		1205 ± 21	2506 ± 77	3223 ± 794	3802.5 ± 1373.5
ORP, mV		47 ± 11	40.5 ± 9.5	46.5 ± 4.5	46.5 ± 3.5

**Table 7 sensors-25-07548-t007:** Main physicochemical characteristics of biscuit dough with purslane. All data have a statistically significant difference at *p* < 0.05.

	% Additive	0	5	10	15
Characteristic					
pH		6.74 ± 0.04	7.355 ± 0.07	7.34 ± 0	7.43 ± 0.04
TDS, ppm		1877 ± 40	2058.5 ± 153.5	2264 ± 359	2752.5 ± 129.5
EC, µS/cm		3762.5 ± 56.5	4132.5 ± 321.5	4528.5 ± 717.5	5519 ± 273
ORP, mV		31 ± 2	29.5 ± 3.5	27.5 ± 1.5	21.5 ± 0.5

**Table 8 sensors-25-07548-t008:** Values of color indices of biscuit dough with additive purslane flour. All data have a statistically significant difference at *p* < 0.05.

	% Additive	0	5	10	15
Color Index					
C1		110.73 ± 0.49	86.43 ± 0.43	69.17 ± 0.4	70.21 ± 0.25
C2		144.07 ± 0.79	117.5 ± 2.02	78.1 ± 0.84	57.7 ± 0.52
C3		88.73 ± 0.96	84.17 ± 1.6	59.25 ± 0.83	53.98 ± 0.51
C4		137.4 ± 1.93	145.99 ± 2.54	115.98 ± 1.65	106.49 ± 1.02

**Table 9 sensors-25-07548-t009:** Values of spectral indices of biscuit dough with the additive purslane flour. All data have a statistically significant difference at *p* < 0.05.

	% Additive	0	5	10	15
Spectral Index					
S1		0.66 ± 0.02	0.62 ± 0.01	0.57 ± 0.02	0.58 ± 0.02
S2		0.51 ± 0.03	0.55 ± 0.03	0.92 ± 0.02	1.16 ± 0.01
S3		0.21 ± 0.01	0.23 ± 0.02	0.27 ± 0.01	0.27 ± 0.02

**Table 10 sensors-25-07548-t010:** Elemental composition of biscuits with purslane flour. All data have a statistically significant difference at *p* < 0.05.

	% Additive	0	5	10	15
Element					
Ca, mg/kg		950.53 ± 4.62	1001.48 ± 1.8	1272.53 ± 2.45	1523.51 ± 2.53
Cu, mg/kg		2.8 ± 0.21	4.01 ± 0.2	4.8 ± 0.23	5.91 ± 0.19
Fe, mg/kg		40.81 ± 0.37	65.22 ± 0.3	61.57 ± 0.26	70.13 ± 0.24
K, mg/kg		1434.85 ± 4.23	3372.94 ± 2.94	4687.33 ± 6.66	5187.78 ± 7.63
Mg, mg/kg		287.97 ± 5.34	557.93 ± 2.08	685.08 ± 3.03	764.46 ± 1.41
Mn, mg/kg		4.86 ± 0.19	6.75 ± 0.26	7.65 ± 0.26	8.73 ± 0.26
P, %		0.15 ± 0.01	0.15 ± 0.01	0.17 ± 0.01	0.17 ± 0.01
Zn, mg/kg		9.42 ± 0.21	10.56 ± 0.2	11.7 ± 0.21	12.22 ± 0.2

**Table 11 sensors-25-07548-t011:** Chemical analysis of biscuits with purslane flour. All data have a statistically significant difference at *p* < 0.05.

	% Additive	0	5	10	15
Characteristic, %					
Moisture		3.91 ± 003	3.47 ± 0.05	3.56 ± 0.06	3.29 ± 0.02
Dry matter		96.09 ± 0.2	96.53 ± 0.2	96.44 ± 0.3	96.71 ± 0.2
Crude proteins		10.23 ± 0.07	10.36 ± 0.08	10.72 ± 0.1	10.51 ± 0.17
Crude fats		30.93 ± 0.42	31.65 ± 0.42	32.44 ± 0.42	31.47 ± 0.42
Crude fibers		19.79 ± 0.45	19.15 ± 0.49	22.67 ± 0.46	23.77 ± 0.51
Crude ash		1.33 ± 0.03	2.27 ± 0.04	2.95 ± 0.03	3.85 ± 0.04
Nitrogen-free extractives		33.81 ± 0.4	33.1 ± 0.2	27.66 ± 0.3	27.11 ± 0.4

**Table 12 sensors-25-07548-t012:** Main physicochemical characteristics of biscuits with purslane flour. All data have a statistically significant difference at *p* < 0.05.

	% Additive	0	5	10	15
Characteristic					
pH		7.18 ± 0.06	7.29 ± 0.06	7.28 ± 0.08	7.37 ± 0.02
TDS, ppm		1904 ± 30	2089.5 ± 215.5	2285.5 ± 411.5	2429.5 ± 555.5
EC, µS/cm		3801 ± 53	4174.5 ± 426.5	4611.5 ± 863.5	4816.5 ± 1068.5
ORP, mV		70.5 ± 16.5	51.5 ± 2.5	56.5 ± 1.5	33.5 ± 0.5

**Table 13 sensors-25-07548-t013:** Geometric properties and thermal losses of butter biscuits with additive purslane flour. All data show statistically significant differences at *p* < 0.05.

	% Additive	0	5	10	15
Characteristic					
D, mm		46.99 ± 14.87	48.25 ± 15.27	47.47 ± 15.02	47.67 ± 15.08
h, mm		7.62 ± 2.43	7.13 ± 2.31	6.76 ± 2.2	6.45 ± 2.05
SF		6.18 ± 0.29	6.8 ± 0.53	7.06 ± 0.54	7.4 ± 0.27
TL, %		20 ± 0.2	21 ± 0.21	22 ± 0.23	22 ± 0.22

**Table 14 sensors-25-07548-t014:** Sensory evaluation of butter biscuits with additive purslane flour. All data have statistically significant differences at *p* < 0.05.

	% Additive	0	5	10	15
Characteristic					
General appearance		5 ± 0	4.5 ± 0.58	3.5 ± 1	3.75 ± 0.5
Consistency		5 ± 0	4.5 ± 0.58	3.5 ± 1	3.5 ± 0.58
Aroma		5 ± 0	4.75 ± 0.5	3.75 ± 1.26	3.75 ± 0.5
Taste		5 ± 0	4.25 ± 0.5	3.75 ± 1.5	3.25 ± 0.5
Smell		5 ± 0	4.5 ± 0.58	4 ± 1.41	4 ± 0
Chewiness		4.75 ± 0.5	4.25 ± 0.5	3.75 ± 1.26	3.75 ± 0.96
Overall evaluation		4.96 ± 0.08	4.46 ± 0.54	3.71 ± 1.24	3.67 ± 0.51

**Table 15 sensors-25-07548-t015:** Values of color indices of biscuits with additive purslane flour. All data have a statistically significant difference at *p* < 0.05.

	% Additive	0	5	10	15
Color Index					
C1		104.97 ± 0.09	93.25 ± 0.43	88.95 ± 1.65	83.72 ± 0.91
C2		124.53 ± 0.23	118.89 ± 1.3	111.81 ± 0.31	98.58 ± 0.54
C3		75.04 ± 0.27	81.41 ± 0.8	81.45 ± 0.5	76.9 ± 0.77
C4		118.43 ± 0.39	136.85 ± 1.59	139.79 ± 1.93	135.69 ± 1.88

**Table 16 sensors-25-07548-t016:** Values of spectral indices of biscuits with additive purslane flour. All data have statistically significant differences at *p* < 0.05.

	% Additive	0	5	10	15
Spectral Index					
S1		0.62 ± 0.01	0.62 ± 0.01	0.62 ± 0.01	0.61 ± 0.01
S2		0.8 ± 0.01	0.64 ± 0.02	0.66 ± 0.02	0.75 ± 0.02
S3		0.24 ± 0.01	0.23 ± 0.01	0.24 ± 0.01	0.24 ± 0.01

**Table 17 sensors-25-07548-t017:** A comparative analysis between data from experimental (real) biscuits and literature data.

Feature (Experimental)	Meaning	PC1	PC2	Feature (Literature)	Meaning	PC1	PC2
F33	General appearance	−0.53	0.2	FL1	Appearance	−0.41	−0.35
F38	Chewiness	−0.57	0.02	FL2	Chewing resistance	−0.3	−0.17
F34	Aroma	−0.58	0.23	FL4	Odor	−0.44	−0.17
F35	Taste	−0.55	0.16	FL6	Taste	−0.36	0.57
F36	Smell	−0.51	0.29	FL4	Odor	−0.44	−0.17
F39	Overall acceptance	−0.57	0.14	FL5	Overall acceptability	−0.26	−0.37
F46	Protein	−0.31	−0.39	FL15	Protein	0.48	−0.53
F48	Moisture	—	—	FL14	Moisture	−0.36	0.57
F51	Crude fiber	−0.18	−0.36	FL11	Crude fiber	−0.35	−0.27

F—feature of experimental (real) biscuits; FL—feature obtained from literature data.

## Data Availability

The data is presented within the paper.

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
