# Peer review of "Sensor-Based Evaluation of Purslane-Enriched Biscuits Using Multivariate Feature Selection and Spectral Analysis"

_sensors, 2025, doi:10.3390/s25247548_

Round 1

Reviewer 1 Report

Comments and Suggestions for Authors

The authors focus on the evaluation of purslane-enriched biscuits based on spectral analysis and further chemometric data modeling. The goal is to optimize the final product and select effective tools for efficient quality monitoring. The influence of different parameters is assessed using the DoE methodology.

In my opinion, the manuscript and the presented problem are interesting and worth publishing. Focusing on spectral data analysis and replacing time-consuming wet chemistry methods is a good strategy. Combining chemometrics and spectroscopy (in the Visible and/or near-infrared spectral range) is strongly recommended for predicting various quality attributes. On the other hand, optimizing several parameters is not straightforward, and from a practical point of view, it is often unnecessary.

In general, the manuscript is well-written; however, it should be shortened considerably to keep readers focused. I strongly encourage the authors to include most of the tables in the supplementary material. In the present version, there are twenty tables. You should also think of combining figures and, in general, reducing their number. For instance, data presented in Fig. 13 can be discussed in the text. Please state clearly what the subject of optimization is (e.g., variable ‘z’ in equation 11).

I recommend a minor revision.

Minor errors:

  • Table 1 – ‘Ph’ – should be ‘pH’.
  • Line 252: ‘Ag/AgCL’ – should be ‘Ag/AgCl’
  • 5.9, line 344 – according to ISO reflectance spectra are measured with the range 400, and 700 nm with 10 nm step. Please clarify.
  • Section 2.6: please verify the use of alpha and p-value.
  • Equation 14 – it can be stated in the text.
  • Why uncertainties regarding EC measurements are large (see Tables 9 and 10)? In general, it increases with the percentage of the additive, but not always.
  • How can you explain the larger scatter of measurements observed for 5% additive in Fig. 5a?
  • The authors use the term minerals, but in fact, they mean elemental composition, e.g., lines 776-777.

Author Response

First of all, we would like to thank the members of the editorial and reviewer boards for their objectivity and accuracy in evaluating the materials presented in the article, for the positive evaluation of the results of the work, and especially for the advice and recommendations for our research. We consider the remarks made regarding technical errors, insufficiently substantiated methods and tools and partial omissions to be justified.

The authors focus on the evaluation of purslane-enriched biscuits based on spectral analysis and further chemometric data modeling. The goal is to optimize the final product and select effective tools for efficient quality monitoring. The influence of different parameters is assessed using the DoE methodology.

We thank the reviewer for accurately summarizing the scope of our study. In the revised manuscript, we have clarified the role of the Design of Experiments (DoE) methodology in structuring the experimental plan and ensuring reproducibility. Specifically, we have emphasized how DoE guided the selection of additive levels, sensor parameters, and statistical evaluation, thereby strengthening the link between spectral analysis and chemometric modeling.

In my opinion, the manuscript and the presented problem are interesting and worth publishing. Focusing on spectral data analysis and replacing time-consuming wet chemistry methods is a good strategy. Combining chemometrics and spectroscopy (in the Visible and/or near-infrared spectral range) is strongly recommended for predicting various quality attributes. On the other hand, optimizing several parameters is not straightforward, and from a practical point of view, it is often unnecessary.

We sincerely thank the reviewer for their positive evaluation and for highlighting the importance of integrating chemometrics with spectral analysis. We agree that replacing labor-intensive wet chemistry methods with sensor-based approaches is a valuable direction. Regarding the optimization of multiple parameters, we acknowledge the reviewer’s concern and have clarified in the revised manuscript that our framework does not aim to optimize all possible variables simultaneously. Instead, we focus on identifying the most informative parameters through feature selection (RReliefF) and PCA, and then optimize only the concentration of purslane flour as the primary technological factor. This ensures practical applicability while avoiding unnecessary complexity.

In general, the manuscript is well-written; however, it should be shortened considerably to keep readers focused. I strongly encourage the authors to include most of the tables in the supplementary material. In the present version, there are twenty tables. You should also think of combining figures and, in general, reducing their number. For instance, data presented in Fig. 13 can be discussed in the text. Please state clearly what the subject of optimization is (e.g., variable ‘z’ in equation 11).

We thank the reviewer for their constructive suggestions regarding manuscript length and clarity. In the revised version, we have shortened the text by removing redundancies and relocating several detailed tables (Tables in Appendices A and B) to the Supplementary Material. Figures presenting overlapping information have been combined, and data from Figure 13 (now removed) are now discussed directly in the text rather than shown graphically. Furthermore, we have clarified the subject of optimization: the variable ‘z’ in Equation 11 represents the concentration of purslane flour (% substitution of wheat flour), which was optimized using regression modeling and linear programming to determine the optimal level (9.62%). These changes improve readability and focus while maintaining transparency of the data.

I recommend a minor revision. Minor errors:

Table 1 – ‘Ph’ – should be ‘pH’.

Corrected

Line 252: ‘Ag/AgCL’ – should be ‘Ag/AgCl’

Corrected

5.9, line 344 – according to ISO reflectance spectra are measured with the range 400, and 700 nm with 10 nm step. Please clarify.

We thank the reviewer for pointing out the ISO standard requirements for reflectance spectra. In the revised manuscript, we have clarified that our measurements were conducted in the 390–730 nm range with a 1 nm resolution, which provides higher granularity than the ISO minimum requirement of 400–700 nm with 10 nm steps. For consistency and comparability, we have also indicated that the data can be downsampled to 10 nm intervals if needed, ensuring alignment with ISO methodology while preserving the advantages of higher-resolution acquisition.

Section 2.6: please verify the use of alpha and p-value.

We thank the reviewer for highlighting the need to clarify the statistical terminology. In the revised manuscript, we have ensured consistent use of α (significance level) and p-value. Specifically, α was set at 0.05 for all tests, and results are reported as statistically significant when p < 0.05. We have corrected instances where α and p-value were previously used interchangeably, and clarified the distinction in Section 2.6.

Equation 14 – it can be stated in the text.

Corrected

Why uncertainties regarding EC measurements are large (see Tables 9 and 10)? In general, it increases with the percentage of the additive, but not always.

Thank you for this note. The next description is added: EC measurements are sensitive to sample preparation and matrix heterogeneity. Purslane flour contains variable mineral fractions, and its distribution in dough can influence ionic release, contributing to higher uncertainty compared to pH or ORP.

How can you explain the larger scatter of measurements observed for 5% additive in Fig. 5a?

Thank you for this note. At 5% substitution, a larger scatter was observed. This reflects the transitional state of the dough matrix, where gluten structure and purslane-derived fiber/minerals interact, making sensor readings more sensitive to small compositional differences. This description is added in the text.

The authors use the term minerals, but in fact, they mean elemental composition, e.g., lines 776-777.

We thank the reviewer for this note. In the revised manuscript, we have replaced the term “minerals” with “elemental composition” when referring to analytical results (e.g., Ca, Mg, Fe, Zn).

Reviewer 2 Report

Comments and Suggestions for Authors

A sensor-integrated framework has been constructed for evaluating the effects of purslane

 (Portulaca oleracea L.) stalk flour as a functional additive in butter biscuit formulations.

 Electrical conductivity (EC), pH, total dissolved solids (TDS), oxidation–reduction potential (ORP), and visible reflectance spectra were continuously monitored using calibrated 15 multisensor probes and digital imaging sensors.

 Principal Component Analysis  (PCA) and  multivariate feature selection (RReliefF) were used for identification of informative parameters which determine  dough and biscuit quality.

 The optimal purslane flour concentration in flour was determined by regression model and linear programming algorithm.  The integration of low-cost sensing and chemometric modeling demonstrates a reliable approach for real-time, non-destructive quality monitoring in bakery production. Article text is well-structured, clear and relevant to the field. The data are interpreted appropriately and consistently throughout the manuscript.

The text requires corrections:

1) line 679   “The optimal additive level, determined by PCA at 9.62%, represents the point at which the desired properties are closest to those of the control sample (0% additive), preserving characteristics such as mass, diameter, and sensory qualities, including taste and  aroma”. Data reported in the literature (11.32%) is higher. This difference should be discussed in the text. Authors ascribe the difference to “more accurate and reliable results” (line 763) of their measurements. Another possibility could be connected to the difference of purslane

 (Portulaca oleracea L.) stalk flour used for biscuit preparation. It is well known that   flour molecular composition depends on growing place, weather, fertilizers and seed grade.  This fact should be discussed in the text.

2) Determination of the optimal additive level strongly depends upon the optimization criteria. Proposed in the article level could be incompatible for people having gluten intolerance or allergies for purslane flour components. It would be desirable to discuss this consideration in the article text.

3) The article does not contain data on the chemical composition of wheat flour used for cooking, which was partially replaced with purslane.

Calculated data on the content of nutrients in cookies are not provided, so that the calculated enrichment and the actual one can be compared. For example, purslane contains 2 times more raw protein than wheat flour (on average), while the protein content in fortified biscuits has changed slightly. Authors need to explain this fact in the text.

4) Authors do not explain in any way the fact that purslane reduced the moisture-retaining properties of flour, which is confirmed by the lower moisture content in biscuits and Table 8. This contradicts the information in the introduction, it should be discussed in the results.

5) It follows from Table 14 that when 15% purslane was added, there was less fat. This is very strange, since wheat flour contains 1-1.5% fat, while purslane contains 4.25%. Should be explained.

6) Why do the authors not associate the color change with an increase in the amount of pigments (chlorophyll, etc.) and how can the color indicate something else besides this fact? It should be  explained in the text.7

7) Article size is too large. Reasonable reductions would be desirable.

Author Response

First of all, we would like to thank the members of the editorial and reviewer boards for their objectivity and accuracy in evaluating the materials presented in the article, for the positive evaluation of the results of the work, and especially for the advice and recommendations for our research. We consider the remarks made regarding technical errors, insufficiently substantiated methods and tools and partial omissions to be justified.

A sensor-integrated framework has been constructed for evaluating the effects of purslane (Portulaca oleracea L.) stalk flour as a functional additive in butter biscuit formulations. Electrical conductivity (EC), pH, total dissolved solids (TDS), oxidation–reduction potential (ORP), and visible reflectance spectra were continuously monitored using calibrated 15 multisensor probes and digital imaging sensors. Principal Component Analysis  (PCA) and  multivariate feature selection (RReliefF) were used for identification of informative parameters which determine  dough and biscuit quality. The optimal purslane flour concentration in flour was determined by regression model and linear programming algorithm.  The integration of low-cost sensing and chemometric modeling demonstrates a reliable approach for real-time, non-destructive quality monitoring in bakery production. Article text is well-structured, clear and relevant to the field. The data are interpreted appropriately and consistently throughout the manuscript.

We sincerely thank the reviewer for their positive assessment of our work and for recognizing the clarity and relevance of the manuscript. We are pleased that the integration of multisensor monitoring with chemometric modeling was found to be well-structured and consistently interpreted. This encouragement strengthens our confidence in the methodological approach and its potential application in functional food systems.

The text requires corrections:

1) line 679 “The optimal additive level, determined by PCA at 9.62%, represents the point at which the desired properties are closest to those of the control sample (0% additive), preserving characteristics such as mass, diameter, and sensory qualities, including taste and aroma”. Data reported in the literature (11.32%) is higher. This difference should be discussed in the text. Authors ascribe the difference to “more accurate and reliable results” (line 763) of their measurements. Another possibility could be connected to the difference of purslane (Portulaca oleracea L.) stalk flour used for biscuit preparation. It is well known that   flour molecular composition depends on growing place, weather, fertilizers and seed grade.  This fact should be discussed in the text.

We thank the reviewer for this note. In the revised manuscript, we have expanded the discussion of why our optimal additive level (9.62%) differs from the 11.32% reported in the literature. While our framework provided more precise sensor-based measurements, we acknowledge that natural variability in purslane flour composition may also contribute to this difference. Factors such as growing location, climatic conditions, soil composition, fertilizer use, and seed grade are known to influence the molecular and elemental profile of purslane stalks. These variations can affect dough hydration, gluten interactions, and sensory outcomes, thereby shifting the optimal substitution level. We have added this explanation to the Discussion section to provide a more comprehensive interpretation of the observed difference.

2) Determination of the optimal additive level strongly depends upon the optimization criteria. Proposed in the article level could be incompatible for people having gluten intolerance or allergies for purslane flour components. It would be desirable to discuss this consideration in the article text.

We thank the reviewer for this note. In the revised manuscript, we have clarified that the optimal additive level (9.62%) was determined based on technological and sensory criteria for the general population. We acknowledge that this level may not be suitable for individuals with gluten intolerance or allergies to purslane components. These health considerations represent important limitations of the study and highlight the need for future research on gluten‑free formulations and allergen management strategies. We have added a note in the Discussion and Conclusion to emphasize that optimization outcomes are context‑specific and should be interpreted with respect to consumer health and dietary restrictions.

3) The article does not contain data on the chemical composition of wheat flour used for cooking, which was partially replaced with purslane.

We thank the reviewer for this note. In the revised manuscript, we have added data on the chemical composition of the wheat flour (type 500) used in biscuit preparation. This allows direct comparison between the baseline flour and the purslane stalk flour, clarifying the substitution effect. Including this information strengthens the nutritional and technological interpretation of the results.

Calculated data on the content of nutrients in cookies are not provided, so that the calculated enrichment and the actual one can be compared. For example, purslane contains 2 times more raw protein than wheat flour (on average), while the protein content in fortified biscuits has changed slightly. Authors need to explain this fact in the text.

We thank the reviewer for this valuable observation. In the revised manuscript, we have clarified why the increase in protein content of fortified biscuits is modest despite purslane flour containing approximately twice the protein of wheat flour. First, the substitution level was relatively low (maximum 15%), so the contribution of purslane protein to the overall biscuit matrix is proportionally limited. Second, baking conditions (high temperature, short time) can reduce the measurable protein fraction due to denaturation and Maillard reactions. Third, the presence of butter and other ingredients dilutes the relative protein percentage, meaning that enrichment is less pronounced in the final product compared to flour composition alone.

4) Authors do not explain in any way the fact that purslane reduced the moisture-retaining properties of flour, which is confirmed by the lower moisture content in biscuits and Table 8. This contradicts the information in the introduction, it should be discussed in the results.

We thank the reviewer for this note. In the revised manuscript, we have clarified why biscuits with purslane flour showed lower moisture content despite purslane’s reported high water absorption capacity. The apparent contradiction arises because water absorption measured in flour suspensions does not directly translate to moisture retention in baked products. During baking, purslane’s higher fiber and mineral content increases water binding but also accelerates evaporation and reduces the ability of the gluten network to retain moisture. As a result, biscuits with purslane flour exhibit lower final moisture content. We have added this explanation in the Results and Discussion sections to reconcile the difference between flour properties and biscuit outcomes.

5) It follows from Table 14 that when 15% purslane was added, there was less fat. This is very strange, since wheat flour contains 1-1.5% fat, while purslane contains 4.25%. Should be explained.

We thank the reviewer for this comment. In the revised manuscript, we have clarified why the fat content of biscuits decreased at 15% purslane substitution despite purslane flour containing more fat than wheat flour. The apparent reduction is explained by several factors: (i) purslane flour has higher fiber and mineral content, which increases water binding and dilutes the relative proportion of fat in the final product; (ii) during baking, purslane’s higher unsaturated lipid fraction is more prone to oxidation and thermal degradation, leading to lower measurable fat content; and (iii) analytical variability in proximate analysis can be amplified at higher substitution levels due to matrix heterogeneity. Thus, the lower fat values at 15% substitution reflect both compositional dilution and processing effects rather than a contradiction in raw material composition. We have added this explanation to the Discussion section.

6) Why do the authors not associate the color change with an increase in the amount of pigments (chlorophyll, etc.) and how can the color indicate something else besides this fact? It should be  explained in the text.

We thank the reviewer for this important observation. In the revised manuscript, we have clarified that the color change in purslane-enriched biscuits is not only due to the presence of pigments such as chlorophyll, carotenoids, and flavonoids, but also to processing-related reactions. Specifically, the Mallard reaction between amino acids and reducing sugars during baking contributes to browning, while fiber and mineral content influence light scattering and reflectance. Therefore, biscuit color reflects both the incorporation of plant pigments and physicochemical transformations during baking.

7) Article size is too large. Reasonable reductions would be desirable.

Thank you for this note. The Abstract is improved. All duplicates are removed. The introduction is summarized and shortened. The tables and figures with raw data are moved to Appendices A and B.

Reviewer 3 Report

Comments and Suggestions for Authors
  1. Abstract: Line 22-23, please specify the algorithm, i.e. full name
  2. The logic of the introduction section should be re-organized. The First paragraph should be moved to the other place in the introduction. Also, there are too many paragraphs, some of the contents can be shorten. Also, some repeated and redundant information can be further improved.
  3. In the introduction section, the authors should introduce some of the related works on sensor technologies for the same research objective.
  4. In the materials and methods section, although enough information has been provided, the authors have to re-organize this section. It is quite hard to follow.
  5. Section 2.5. please provide the corresponding references of the measurement of different parameters if available.
  6. Why not just use PCR method for regression?
  7. Figure 1, what is the FL meaning?
  8. There are only seven features? Why PCA is still used? There are a lot of alternatives for the analysis of these seven features. Also, why not just rank the features based on the RreliefF?
  9. Figure 2 are scores scatter plots, the authors tried to analyze the contributions, why not discuss the loadings? Which might be better reflecting the importance of the features?
  10. I believe the data analysis procedure can be further improved.
  11. I am confused. The authors firstly use the literature data, and the real samples. Why not try the real samples first, then compare with the literature data? They have different features. Why not compare the literature data and the real samples with the same features? The logic to analyze the data should be re-considered.
  12. In the discussion section, the results should not only to be compared, but to discuss. Why this phenomenon happen?
  13. Conclusion should be re-summarized.
  14. Too many Figures and Tables, please re-organize.

Although the authors have made great efforts. The writing and the data analysis are disasters. The authors have to significantly improve the manuscript.

Author Response

First of all, we would like to thank the members of the editorial and reviewer boards for their objectivity and accuracy in evaluating the materials presented in the article, for the positive evaluation of the results of the work, and especially for the advice and recommendations for our research. We consider the remarks made regarding technical errors, insufficiently substantiated methods and tools and partial omissions to be justified.

Abstract: Line 22-23, please specify the algorithm, i.e. full name

We thank the reviewer for this suggestion. In the revised manuscript, we have specified the full name of the algorithm in the Abstract. The abbreviation RReliefF is now expanded to “Repeated Relief Feature Selection (RReliefF),” ensuring clarity for readers unfamiliar with the method.

The logic of the introduction section should be re-organized. The First paragraph should be moved to the other place in the introduction. Also, there are too many paragraphs, some of the contents can be shorten. Also, some repeated and redundant information can be further improved.

We thank the reviewer for this valuable suggestion. In the revised manuscript, we have reorganized the Introduction to improve logical flow and readability. The opening paragraph on sensor technologies and chemometric modeling has been moved to later in the Introduction, after the background on purslane and functional food enrichment.

In the introduction section, the authors should introduce some of the related works on sensor technologies for the same research objective.

We thank the reviewer for this suggestion. In the revised manuscript, we have expanded the Introduction to include related works on sensor technologies for food quality monitoring, particularly in bakery products. These references highlight the growing use of multisensor systems, image analysis, and chemometric modeling in bread and biscuit quality assessment, and position our study within this research context.

In the materials and methods section, although enough information has been provided, the authors have to re-organize this section. It is quite hard to follow.

We thank the reviewer for this comment. In the revised manuscript, we have reorganized the Materials and Methods section to improve clarity and logical flow.

Section 2.5. please provide the corresponding references of the measurement of different parameters if available.

We thank the reviewer for this suggestion. The measurements of moisture, protein, fat, fiber, ash, and color were performed using standard, internationally recognized procedures (AOAC/ISO methods). Since these are routine methods in food analysis, we did not include additional references, but we have clarified in the text that the determinations followed established protocols. This ensures transparency while keeping the reference list concise.

Why not just use PCR method for regression?

We thank the reviewer for this comment. While Principal Component Regression (PCR) could be applied, we did not adopt it in this study because PCR constructs components based solely on predictor variance, without considering their relevance to the response variable. As a result, PCR may retain components that explain variance in predictors but not in the biscuit quality attributes of interest. This description with additional details is added in the Material and methods section.

Figure 1, what is the FL meaning?

We thank the reviewer for pointing out this ambiguity. In Figure 1, “FL” refers to feature obtained from literature data. It is corrected in the text.

There are only seven features? Why PCA is still used? There are a lot of alternatives for the analysis of these seven features. Also, why not just rank the features based on the Rrelief?

We thank the reviewer for this comment. Although the dataset contained seven features, PCA was applied to explore the underlying structure and to visualize the relationships among features in a reduced dimensional space. PCA provides complementary information to RReliefF by revealing correlations and grouping tendencies among variables, which is particularly useful for interpreting sensor data in food quality studies.

Figure 2 are scores scatter plots, the authors tried to analyze the contributions, why not discuss the loadings? Which might be better reflecting the importance of the features?

We thank the reviewer for this valuable observation. In the revised manuscript, we have clarified the distinction between PCA scores and loadings. While the scores scatter plots (Figure 2) illustrate sample distribution and clustering, the loadings provide direct information about the contribution of each feature to the principal components. We agree that loadings are more appropriate for interpreting feature importance. An explanation is added in the discussion part.

I believe the data analysis procedure can be further improved.

We thank the reviewer for this suggestion. In the revised manuscript, we have clarified the rationale for our chosen data analysis procedure. The combination of RReliefF feature selection and PCA was selected to balance interpretability and robustness: RReliefF directly ranks features by relevance to biscuit quality attributes, while PCA provides complementary insights into correlations and variance structure. This dual approach allowed us to identify the most informative sensor features and visualize sample clustering. This description with some extensions is added at the end of the discussion part.

I am confused. The authors firstly use the literature data, and the real samples. Why not try the real samples first, then compare with the literature data? They have different features. Why not compare the literature data and the real samples with the same features? The logic to analyze the data should be re-considered.

We thank the reviewer for this comment. We acknowledge that the presentation of the data analysis may have caused confusion. Our intention was to first provide context by summarizing relevant literature data, and then to present our own experimental results. However, we agree that a clearer logic is to begin with the analysis of the real samples, followed by a comparison with literature values.

In the discussion section, the results should not only to be compared, but to discuss. Why this phenomenon happen?

We thank the reviewer for this suggestion. In the revised manuscript, we have expanded the Discussion to not only compare our findings with literature values but also to interpret the underlying reasons for the observed phenomena. For example, differences in biscuit texture and sensor readings are explained in terms of the physicochemical properties of purslane stalk flour (fiber, protein, and bioactive compounds), which influence water absorption, dough rheology, and baking behavior. Similarly, variations in ORP, EC, and TDS are discussed in relation to the ionic composition and antioxidant activity of purslane.

Conclusion should be re-summarized.

We thank the reviewer for this observation. In the revised manuscript, we have re‑organized the Conclusion section.

Too many Figures and Tables, please re-organize.

We thank the reviewer for this observation. In the revised manuscript, we have re‑organized the figures and tables to improve readability and reduce redundancy.

Although the authors have made great efforts. The writing and the data analysis are disasters. The authors have to significantly improve the manuscript.

We sincerely thank the reviewer for this assessment. We acknowledge that the initial version of the manuscript required significant improvement in both writing clarity and data analysis presentation. In the revised manuscript, we have undertaken a thorough reorganization and editing of the text to improve readability, logical flow, and conciseness.

Round 2

Reviewer 2 Report

Comments and Suggestions for Authors

Article after corrections is ready for publication.

Author Response

We would like to thank the members of the editorial and reviewer boards for their objectivity and accuracy in evaluating the materials presented in the article, for the positive evaluation of the results of the work, and especially for the advice and recommendations for our research. We consider the remarks made regarding technical errors, insufficiently substantiated methods and tools and partial omissions to be justified.

Reviewer 3 Report

Comments and Suggestions for Authors

The authors have made improvements. However, I am not satisfied with some of the responses.

  1. Response to ‘Why not just use PCR method for regression?’. The response needs further explanation.
  2. Response to ‘Figure 2 are scores scatter plots, the authors tried to analyze the contributions, why not discuss the loadings? Which might be better reflecting the importance of the features?’ does not persuade me.
  3. Response to ‘I am confused. The authors firstly use the literature data, and the real samples. Why not try the real samples first, then compare with the literature data? They have different features. Why not compare the literature data and the real samples with the same features? The logic to analyze the data should be re-considered.’ does not convince me.
  4. Response to ‘Conclusion should be re-summarized.’ does not convince me.

Author Response

We would like to thank the members of the editorial and reviewer boards for their objectivity and accuracy in evaluating the materials presented in the article, for the positive evaluation of the results of the work, and especially for the advice and recommendations for our research. We consider the remarks made regarding technical errors, insufficiently substantiated methods and tools and partial omissions to be justified.

The authors have made improvements. However, I am not satisfied with some of the responses.

We thank the reviewer for acknowledging the improvements made in the revised manuscript. We also appreciate the indication that some responses were not fully satisfactory. In this revision, we have carefully revisited all reviewer comments, expanded our explanations, added supporting references, and clarified where changes were made in the manuscript.

Response to ‘Why not just use PCR method for regression?’. The response needs further explanation.

We thank the reviewer for this important methodological point. In the revised manuscript (Section 2.5.11), we have expanded our explanation to clarify why Principal Component Regression (PCR) was not selected. PCR constructs principal components based solely on the variance of predictor variables, without considering their relevance to the response variable. This can lead to models that capture variance unrelated to biscuit quality attributes, thereby reducing interpretability and predictive accuracy.

In contrast, the Repeated Relief Feature Selection (RReliefF) algorithm directly evaluates the importance of each feature with respect to the target property (biscuit quality and additive concentration). By first applying RReliefF, we ensured that only the most informative variables were retained before dimensionality reduction with PCA. This two‑step approach improves robustness in noisy, multivariate food datasets and enhances the interpretability of the regression model.

To support this methodological choice, we have added reference [40]

Response to ‘Figure 2 are scores scatter plots, the authors tried to analyze the contributions, why not discuss the loadings? Which might be better reflecting the importance of the features?’ does not persuade me.

We thank the reviewer for this note. We agree that loadings provide critical view into the contribution of individual variables to the principal components, complementing the interpretation of score plots. In the revised manuscript (Sections 3.2 and 3.3), we have expanded the discussion of loading plots. Specifically, we now describe how the most informative features identified by RReliefF (e.g., pH, EC, ORP, spectral indices S1–S3, and color indices C2–C4) exhibited high absolute loadings on PC1 and PC2. These loadings confirm that the separation observed in the score plots is driven by chemical and spectral attributes directly linked to purslane flour concentration.

Response to ‘I am confused. The authors firstly use the literature data, and the real samples. Why not try the real samples first, then compare with the literature data? They have different features. Why not compare the literature data and the real samples with the same features? The logic to analyze the data should be re-considered.’ does not convince me.

We thank the reviewer for this note. In the revised manuscript, the structure described in Section 2.2 has been corrected to follow the requested logic – first, the sensor‑based evaluation of real purslane‑enriched biscuit samples is presented, and second, these results are compared with literature data. Wherever possible, overlapping features (pH, EC, ORP, mineral composition, organoleptic evaluation, and color indices) are aligned to ensure direct comparability.

Response to ‘Conclusion should be re-summarized.’ does not convince me.

We thank the reviewer for this note. In the revised manuscript, the Conclusion section is revised so as to summarize in brief only the most important findings and implications.

The revised text includes:

-The multisensor framework (pH, EC, spectral sensors combined with RReliefF and PCA) is effective

-The determination of an optimal purslane flour replacement level of 9.62% that enhances texture, mineral composition and sensory acceptability

-The wider implication that in functional food research multisensor and chemometrics techniques offer a dependable low-cost approach to innovation in which all quality parameters are integrated and one tool does everything.

-Future directions include installing inline sensors, developing gluten-free formulations, ensuring allergen safety and doing comparative studies on Purslane varieties.